# Effectiveness of Bioactive Compound as Antibacterial and Anti-Quorum Sensing Agent from *Myrmecodia pendans*: An In Silico Study

**DOI:** 10.3390/molecules26092465

**Published:** 2021-04-23

**Authors:** Mieke Hemiawati Satari, Eti Apriyanti, Hendra Dian Adhita Dharsono, Denny Nurdin, Meirina Gartika, Dikdik Kurnia

**Affiliations:** 1Department of Oral Biology, Faculty of Dentistry, Universitas Padjadjaran, Bandung 40132, Indonesia; 2Department of Chemistry, Faculty of Mathematics and Natural Science, Universitas Padjadjaran, Sumedang 45363, Indonesia; eti.apriyanti@unpad.ac.id; 3Department of Conservative Dentistry, Faculty of Dentistry, Universitas Padjadjaran, Bandung 40132, Indonesia; adhita.dharsono@fkg.unpad.ac.id (H.D.A.D.); denny.nurdin@fkg.unpad.ac.id (D.N.); 4Department of Pediatric Dentistry, Faculty of Dentistry, Universitas Padjadjaran, Bandung 40132, Indonesia; meirina.gartika@fkg.unpad.ac.id

**Keywords:** anti-bacteria, molecular docking, *Myrmecodia pendans*, protein, quorum sensing

## Abstract

Background: antibiotic resistance encourages the development of new therapies, or the discovery of novel antibacterial agents. Previous research revealed that *Myrmecodia pendans* (Sarang Semut) contain potential antibacterial agents. However, specific proteins inhibited by them have not yet been identified as either proteins targeted by antibiotics or proteins that have a role in the quorum-sensing system. This study aims to investigate and predict the action mode of antibacterial compounds with specific proteins by following the molecular docking approach. Methods: butein (**1**), biflavonoid (**2**), 3″-methoxyepicatechin-3-*O*-epicatechin (**3**), 2-dodecyl-4-hydroxylbenzaldehyde (**4**), 2-dodecyl-4-hydroxylbenzaldehyde (**5**), pomolic acid (**6**), betulin (**7**), and sitosterol-(6′-*O*-tridecanoil)-3-*O*-*β*-D-glucopyranoside (**8**) from *M. pendans* act as the ligand. Antibiotics or substrates in each protein were used as a positive control. To screen the bioactivity of compounds, ligands were analyzed by Prediction of Activity Spectra for Substances (PASS) program. They were docked with 12 proteins by AutoDock Vina in the PyRx 0.8 software application. Those proteins are penicillin-binding protein (PBP), MurB, Sortase A (SrtA), deoxyribonucleic acid (DNA) gyrase, ribonucleic acid (RNA) polymerase, ribosomal protein, Cytolysin M (ClyM), FsrB, gelatinase binding-activating pheromone (GBAP), and PgrX retrieved from UniProt. The docking results were analyzed by the ProteinsPlus and Discovery Studio software applications. Results: most compounds have Pa value over 0.5 against proteins in the cell wall. In nearly all proteins, biflavonoid (**2**) has the strongest binding affinity. However, compound **2** binds only three residues, so that **2** is the non-competitive inhibitor. Conclusion: compound **2** can be a lead compound for an antibacterial agent in each pathway.

## 1. Introduction

Infectious disease leads to sickness, including infection caused by bacteria [1]. Moreover, pathogenic bacteria can increase the probability of being exposed to other diseases [2]. Antibiotic resistance intensifies the difficulty to deal with pathogenic bacteria, and the development of new therapy or the discovery of novel antibacterial agents plays a vital role to counteract this [3].

Strategic pathways that can be inhibited by antibacterial agents, especially in Gram-positive bacteria, are in the step of the cell wall, protein, ribonucleic acid (RNA), and deoxyribonucleic acid(DNA) synthesis. The processes of their syntheses involve many proteins that support them. In cell wall bacteria, the MurB enzyme contributes to the first step of peptidoglycan synthesis, while penicillin-binding proteins (PBPs) play a role in the final step of cell wall biosynthesis [4,5]. Moreover, the Sortase A (SrtA) enzyme, a protein anchored to the cell wall, is also an important agent for virulence and biofilm formation [6]. The 30S and 70S ribosomal subunits are targets of antibiotics in protein synthesis [7,8]. DNA gyrase participates in the transcription and replication of DNA, while RNA polymerase has a key role in DNA transcription. Both are involved in the first step of gene expression [9,10]. Those proteins are a key in each position that can be a target-inhibited antibacterial agent.

On the other hand, the virulence factor of bacteria can be pressed by blocking the quorum-sensing systems [11]. The quorum-sensing (QS) system is a system of cell–cell communication of bacteria that controls gene expression. Otherwise, proteins and/or receptors that contribute to QS can be targeted for antibacterial therapy. This is an alternative approach to fight pathogenic bacteria. All bacteria, even if different species, can communicate with each other through the general signal [12]. However, each bacterium has a specific QS system and signal that can be detected only the similar species of bacteria. A specific signal called pheromone from one bacterium and receptor that receives this signal in another bacterium is a key point in the QS system [13]. By blocking signal-receptor binding, the QS system breaks out and the virulence factor cannot be expressed [14,15].

Gram-positive QS pathways are divided into four main groups based on the types of the pheromones and their receptors: (1) members of the Rap, NprR, PlcR, and PrgX (RNPP) family of regulators; (2) Agr-type cyclical pheromones; (3) peptides with double-glycine (Gly–Gly) processing motifs; and (4) regulators of the Rgg family. The *Enterococcus faecalis* is known to at least have first and second types of QS [16]. PrgX, is a member of the RNPP family of regulators that controls the conjugative transfer genes expression of the *E. faecalis* plasmid pCF10 in response to an intercellular peptide pheromone signal [17]. Moreover, the FsrB system is an Agr-type like cyclical pheromone in *E. faecalis* [18], while gelatinase biosynthesis-activating pheromone (GBAP) is a pheromone in this system [19,20]. FsrB system encodes gelE and sprE to express gelatinase and serine protease as virulence factors. Another virulence factor found in isolated *E. faecalis* is cytolysin. There are cytolysin M (ClyM) that become the protein-expressing toxin structural components CyL_S_ and CyL_L_ [21,22,23,24].

Besides the potency of proteins as an antibacterial target, we have to screen candidate antibacterial agents that are suitable for each mechanism. Hence, exploring the antibacterial agent from the natural product as a source bioactive compound must be considered [25]. Sarang semut, *Myrmecodia pendans*, is one of the original Indonesian herb plants known to consist of antibacterial compounds [26,27,28,29,30]. There are flavonoids, phenolics, steroids, and terpenoids, which have the ability to inhibit and kill bacteria including *E. faecalis, Streptococcus mutans, and Streptococcus sanguinis* [31,32,33,34]. However, their mechanism has not yet been found. To determine the potency of the compound, we conducted an in silico study to screen the specific ability of the compound. 

The in silico method is a computational method to predict the binding affinity of a small molecule (ligand as an active compound) candidate to a receptor (protein target) in revolving the affinity and activity of a small molecule. This tool is simple and fast, making it suitable for screening bioactive compounds for an in vitro study [35]. Therefore, this study shows the prediction of antibacterial activity in every mechanism to determine the most effective compound as an antibacterial agent from *M. pendans*.

## 2. Results

### 2.1. Bioactivity Prediction of the M. pendans Compound via Prediction of Activity Spectra for Substances (PASS) Online Analysis

Based on PASS analysis, *M. pendans* compounds have a high enough Pa (probability to be active) value, especially activity relating to microbes, as can be seen in Table 1. Generally, the flavonoid group (compound **1**–**3**) is more dominant compared to the others. Compounds **1**–**8** have 28, 21, 14, 11, 9, 3, 16, and 7 activities. However, all compounds have general activities relating to a microbe at a moderate level (Pa value of 0.5). Almost all compounds act as antifungal agents, except compound **5**. The highest value of compounds is found in the antifungal activity, followed by an antibacterial activity. Meanwhile, in specific mechanisms, the Pa value of the compound varies. Data analysis shows that most compounds are a high value of bacteria cell wall inhibitor (at least 0.59), while in the DNA synthesis pathway, they are just a value of 0.3. In action mode in RNA and the protein synthesis pathway, most compounds showed a low Pa value. The Pi value of the compounds is presented in Table 2. Almost all compounds that have Pa < 0.5 have Pi > 0.05.

### 2.2. Prediction of Bioavailability and Antibacterial Activity of M. pendans through Molecular Interaction with Targeted Proteins

#### 2.2.1. Binding Affinity Analysis of Compounds to Proteins

All proteins were docked with ligands not only the *M. pendans* compound, but also to others, as a positive control. According to the docking results, biflavonoid (**2**) has the highest binding affinity on almost all proteins. Its binding affinity is −11.2 kcal·mol^−1^ of PBP, −11.5 kcal·mol^−1^ of MurB, −8.6 kcal·mol^−1^ of SrtA, −7.6 kcal·mol^−1^ of DNA gyrase, −8.6 kcal·mol^−1^ of RNA polymerase subunit alpha, and −9.0 kcal·mol^−1^ of RNA polymerase subunit beta, −9.4 kcal·mol^−1^ of ribosomal subunit 30S, -6.8 kcal·mol^−1^ of ribosomal subunit 50S, −10.4 kcal·mol^−1^ of ClyM, −7.7 kcal·mol^−1^ of FsrB, −6.9 kcal·mol^−1^ of GBAP, and −8.5 kcal·mol^−1^ of PgrX. In the second place, there is 3”-methoxy epicatechin-3-*O*-epicatechin (**3**), followed by pomolic acid (**6**). All three ligands have a binding affinity higher than positive control in each protein (see Table 3). Binding affinity ligand to ribosomal protein 30S is the weakest. Even compound 4 has a binding affinity of −4.7 kcal·mol^−1^. After that, the second weakest binding affinity ligand is to SrtA enzyme.

#### 2.2.2. Hydrogen Bond and Hydrophobic Contact Analysis of Compounds to Proteins

Almost all *M. pendans* compounds have more hydrogen bonds to proteins in the cell wall than proteins in other parts (DNA, RNA, and/or quorum sensing). Similar to the hydrogen bond, hydrophobic contact of ligands mostly appears in proteins in the cell wall. According to Appendix A, compounds **1** and **3** bonds to PBP with fourteen and seven hydrogen bonds, respectively. Moreover, compound **2** formed nine hydrogen bonds to MurB enzyme. Then, SrtA was bound by compounds **3** and **5** with a total of six hydrogen bonds. All three proteins are in the bacteria cell wall. Based on Appendix A, almost all ligands have the most hydrophobic contact with MurB. The total hydrophobic contacts of compounds **3**, **4**, **5**, **6**, and **7** are 5, 8, 4, 5, and 12, respectively. On the other hand, ligands that have the most hydrogen bond to proteins in RNA and DNA synthesis are compound **3** (seeAppendix A). Compound **3** has four and six hydrogen bonds for RNA polymerase subunit alpha and beta. Moreover, it has five and three hydrogen bonds for ribosomal subunits 30S and 50S. Meanwhile, hydrophobic contact at these proteins is not dominated by one compound. The most hydrophobic contact on RNA polymerase subunit alpha is compound **3** (5 bonds), while for RNA polymerase subunit beta, it is compound **1** (5 bonds). Furthermore, there are seven bonds for compound 8-ribosomal subunit30S and compound **2**-ribosomal subunit 50S (see Appendix A). In quorum sensing proteins, such as ClyM, FsrB, GBAP, and PgrX, the sum of hydrogen bond and hydrophobic contact is lower than the others (see Appendix A). In these proteins, the greatest number of bonds is in ClyM complexes, although there are almost no similar residues in each of the ligand-ClyM complexes. Meanwhile, in PgrX, there are at least three of the same residues in each ligand-PgrX complex.

#### 2.2.3. Prediction of Lipinski’s Rule

Based on the results of Lipinski’s rule (see Table 4), only three compounds (compounds **1**, **4**, and **5**) met this rule, while others have two parameters that do meet it, namely, log P, and Molar Refractivity.

#### 2.2.4. Drug-Likeness Analysis of *M. pendans* Compounds

The drug-likeness profile includes clogP, solubility, and topological polar surface area (TPSA) of *M. pendans* compounds, as can be seen in Table 5. Compound **1**–**5** has a good clogP value. However, only compounds **1** and **3** have good solubility. Moreover, all compounds, except compound **3**, meet the TPSA range of drugs (less than 150). TPSA is a factor contributing to the oral bioavailability of the drug. Furthermore, toxicity analysis shows that almost all *M. pendans* compounds have a low risk of mutagenic, tumorigenic, irritant, and reproductive effective effects. 

#### 2.2.5. Pharmacokinetic Prediction of *M. pendans* Compounds

Overall, the pharmacokinetic properties of *M. pendans* compounds are good (Table 6). The compounds do not penetrate the brain (blood–brain barrier or BBB). Several compounds (**1**, **4**, **5**, and **6**) are not absorbed in the gastrointestinal (GI) tract. Moreover, almost no compounds can disturb the metabolism of drugs. This can be seen in just a few ligands that inhibit cytochrome. 

## 3. Discussion

According to PASS analysis data, *M. pendans* compounds **1**–**8** (see Figure 1) have the potential to inhibit microorganisms with varying specifications. In Table 1, the flavonoid group (**1**–**3**) is more active than phenolic (**4** and **5**) and terpenoid (**6**–**8**) groups. Among the 38 types of activities shown in Table 1, butein (**1**) has a Pa value (to be active) in 75% of the activity, while betulin (**7**) as a representative of the steroid group has almost half the amount of activities. In fact, the phenolic group (**4** and **5**) showed a quarter number of activities. However, in general, all compounds except compound **5** have the highest Pa value in antifungal activity, with an average Pa of 0.5, while antibacterial activity is only seen in compounds **1**–**4**. Meanwhile, in a specific activity, compounds have varying Pa values. Most of the compounds can inhibit proteins in the cell wall biosynthetic pathway (six compounds have pa above 0.5) as a membrane permeability inhibitor, Peptidoglycan glycosyltransferase inhibitor, and UDP-*N*-acetylglucosamine 4-epimerase inhibitor. Half of the compounds can block proteins in the DNA synthesis pathway. In contrast, only compound **5** can inhibit tRNA-pseudouridine synthase I with 0.8 Pa and Tpr proteinase, which is among the proteins in the protein synthesis pathway. In addition, most of the compounds are predicted to be an inhibitor of RNA synthesis with a Pa below 0.5. Many Pa values for compounds fall below 0.5. These values correspond to their Pi (to be inactive) value. Most compounds with a Pa of 0.5 have a Pi value above 0.05. However, they still have a Pa value that is higher than the Pi value. For compounds that have Pi < Pa < 0.5, the probability to be active through the experimental activity will be lower. However, if the prediction is confirmed, the compounds found could prove a parent compound for a new chemical class for the biological activity being examined [36].

In addition, docking analysis shows the binding affinity of the compound to be at least −5 kcal·mol^−1^. In the PBP, MurB, and SrtA, the binding affinity of compounds are higher than that of the other proteins with mean binding affinities of −8.7, −8.4, and −6.7 kcal·mol^−1^, respectively. Meanwhile, the average binding affinity of the compound with ClyM, FsrB, GBAP, and PgrX, which are proteins in the quorum-sensing system, is in the second position with the values of −8.7, −6.5, −5.6, and −7.2 kcal·mol^−1^, respectively. Meanwhile, for other proteins, the average binding affinities for compounds are −7, except for ribosomal subunit 50S. They are −7.3 for DNA gyrase, −7.0 for RNA polymerase subunit alpha, −7.6 for RNA polymerase subunit beta, −7.3 for ribosomal subunit 30S, and −5.7 kcal.mol^−1^ for ribosomal subunit 50S. These data show that compounds **1**–**8** are preferred as cell wall inhibitors. Compound **2** is a ligand with the highest binding affinity for each protein, except for RNA polymerase subunit alpha. For this particular protein, compound **3** is the strongest. Both are flavonoids. 

Based on the attachment position, most of the ligands attach to the same pocket in some proteins, such as PBP, MurA, SrtA, DNA gyrase, and protein subunit 50S. The flavonoid group (**1**–**3**) is marked by green, the phenolic group (**4**–**5**) has a violet strip, and the terpenoid group (**6**–**8**) is colored red. Figure 2 shows that ligand positions on RNA polymerase are scattered. It can be seen from the different types of residues that stick to it. Almost no ligands attach the same residue by either a hydrogen bond or hydrophobic contact. Only two ligands that bind Thr216 via hydrogen bond and Pro7 via hydrophobic contact on RNA polymerase alpha. In the RNA polymerase beta, only Tyr353 residue is bound by more than one ligand. Based on the data analysis, all ligands (**1**–**8**) are non-competitive inhibitors against RNA polymerase alpha and beta.

On the other hand, all ligands (**1**–**8**) are shown in the same pocket in DNA gyrase (see Figure 3). This suggests that almost all ligands attach to the same residues such as Asp75, Glu52, Thr167, and Asn48 via hydrogen bond, and Pro81, Val96, and Ile80 via hydrophobic contact. Meanwhile, in Figure 4**,** we can see that only two ligands are bound to different sites in PBP, MurA, and SrtA. In complexes of PBP-ligands, many similar residues are bound by ligands such as Ser232, Lys197, Asn362, Pro210, and Ile364. Penicillin and carbapenems also stick to these residues. Besides that, almost all ligands bind to the same residues by means of hydrophobic bonds. Pro210, Trp202, Arg143, Val213, phe160, and Lys359 appear in complexes of ligand-PBP. This means that all ligands act as competitive inhibitors, except for compounds **2** and **7,** since they do not have the same attached residues. Additionally, in MurB complexes, only compound **4** did not attach the same residues with glycopeptide and quercetin as the positive control. Tyr139, Arg209, Ser222, Gly137, Trp59, and Arg294 appear in most MurB complexes. In the hydrophobic bond analysis, there are the same residues that are bound together such as Tyr133, Trp59, Phe231, Tyr139, Ala138, Ala136, Val61, His255, and Lys234. Furthermore, in SrtA, some compounds (compounds **2**, **3**, **6**, and **8**) attach to Met56, Ile61, Asp137, and Glu112, which are bound by amoxicillin, curcumin, and cefixime, through hydrogen bonds. There are the similar residues between ligands via hydrogen bonds such as Asn127, Cys187, and Ser59 and hydrophobic contacts, such as Tyr197, Lys195, and Ala139. Amoxicillin and cefixime do not have hydrophobic contact, whereas most *M. pendans* compounds have it. 

However, in quorum sensing proteins, such as ClyM, FsrB, GBAP, and PgrX, ligands bind to various sites. Therefore, almost no similar residue bound *M. pendans* compounds and control ligand, as shown in ClyM, FsrB, and GBAP (Figure 5), in either hydrogen bond or hydrophobic contact. In PgrX, only compound **5** adheres to the same residues as the ligand control. Se111, Glu154, and Ser118 are bound by them via hydrogen bond, whereas no residues are the same, which attach to *M. pendans* compound and control ligands. These include all ligands that are non-competitive inhibitors of ClyM, FsrB, GBAP, and PgrX.

Furthermore, *M. pendans* compounds block the 50S and 30S protein subunits with different types (Figure 6). In the 50S subunit, almost all ligands get together in the same pocket, but not in the subunit 30S. Many of the same ligand-bound residues in the 50S subunit indicate that they are competitive inhibitors, while almost no bound residues indicate they are non-competitive inhibitors. 

Among all ligands, compound **2** is the strongest compound that binds to proteins. This suggests that the bulky structure affects the activity of the compound. Ligands that have a larger structure tend to have a stronger binding affinity. This may likely be because they can replenish the active sites of the protein and attach more residues. The interaction between compound **2** and the target protein can be seen in Figure 7 and Figure 8. However, further analysis is still needed to become a drug, especially an oral drug. 

A good standard of the drug must follow Lipinski’s rule of five. According to the Lipinski rule, the ligand must meet several characteristics, namely, molecular weight < 500 Da, log P value <5, acceptor hydrogen bonds <10, and molar refractivity range between 40 and 130. Ligands with a molecular weight of <500 Da are easier to penetrate the cell membrane than ligands with a molecular weight of >500 Da. The log P value is related to the polarity of ligand in fat, oil, and non-polar solvents. Ligands with log P value >5 will more easily interact through the lipid bilayer on the cell membrane, and are widely distributed in the body. This reduces the sensitivity of ligands to reduce the target molecule and increases the ligand toxicity. The smaller the log P value, the more the ligands tend to dissolve in water and become hydrophobic. The log P value of the ligand should not be negative, because it cannot pass through the lipid bilayer membrane (the number of hydrogen bonds on the donor and acceptor correlates with the biological activity of a ligand or drug) [37]. Based on the results of Lipinski’s prediction, compounds **1**, **4**, and **5** can be accepted, while others cannot. Compounds **2**, **6**, **7,** and **8** have log P of more than 5, which means they are too hydrophilic to easily pass through the lipid bilayer, resulting in low sensitivity to the target molecule and high toxicity. Moreover, it is concluded that compounds **2**, **3,** and **8** could not easily penetrate the lipid layer because their molecular weights are over 500 Dalton.

Hydrophilicity, solubility, and topological polar surface area (TPSA) of compounds are measured by the OSIRIS tool. The hydrophilicity of the compound is established by the clogP value. A higher clogP value indicates lower hydrophilicity and, thus, poor absorption and permeation. Most drugs have clogP value less than 5. Compounds **1**–**5** meet this range, while others do not fit. In other words, compounds **1**–**5** are more easily absorb and permeate compared to others. Meanwhile, a log *S* value indicates solubility; the smaller the log *S* value, the higher the solubility, which will enhance the absorption. Most drugs have a log*S* value greater than −4. Only compounds **1** and **3** are in this range. However, all compounds, except compound **3**, meet in the TPSA range of drugs (less than 150). TPSA indicates the surface belonging to polar atoms in the compound. An increased TPSA is associated with diminished membrane permeability and compounds with higher TPSA were better substrates for p-glycoprotein (responsible for drug efflux from the cell). Thus, comparing the compounds, lower TPSA is more favorable for drug-like properties. TPSA is a factor contributing to the bioavailability of the oral drug. Oral bioavailability is the ability of a drug or other substance to become available to the target tissue after its administration. High oral bioavailability is often an important consideration for developing bioactive molecules as therapeutic agents. The higher the bioavailability, the lower the relative molecular mass, the number of donor and acceptor hydrogen bonds, and TPSA value [38]. 

Toxicity analysis shows that almost all *M. pendans* compounds have a low risk of mutagenic, tumorigenic, irritant, and reproductive effective effects. The only butein shows a high-risk probability in mutagenic properties. Furthermore, it is supported by the result of the pharmacokinetic properties of the SwissADME analysis. In the BOILED-Egg analysis (see Figure 9), all compounds do not penetrate the brain (there are many compounds in the yolk area). Several compounds (**1**, **4**, **5**, and **6**) are not absorbed in the gastrointestinal tract (in the white area). However, compounds **1** and **5** are actively effluxed by PGP, while **4** and **6** are non-substrates of PGP. Ligands **2**, **3,** and **7** are considered non-substrate of PGP, which are not absorbed in the brain and the gastrointestinal tract. Moreover, almost all compounds are non-inhibitor cytochrome, especially CYP1A2, CYP2C19, CYP2C9, CYP2D6, and CYP3A4. Therefore, it is expected that they will not interfere with drug metabolism. However, butein (**1**) is predicted to inhibit the work of CYP1A2, CYP2C9, and CYP3A4.

Compared to *M. pendans* compounds, quercetin and curcumin (as the control positive) meet Lipinski’s rule. Quercetin is not absorbed by the brain (BBB), but is absorbed in the human intestines, with a value of 69.799. It also does not cause toxic effects with a value of Caco-2 0.737. Based on the ADME analysis, quercetin will not interrupt the metabolism of the drug, because it does not interfere with cytochrome P450, except for CYP450 1A2 and CYP450 3A4. Conversely, curcumin penetrates the BBB by inhibiting four CYP450 cytochromes (CYP450 1A2, CYP450 2C9, CYP450 2D6, and CYP450 2C19) and human gastrointestinal absorption (HIA). Moreover, it can be toxic [39,40]. 

## 4. Materials and Methods 

### 4.1. Materials

We used twelve target proteins that were retrieved from Protein Data Bank (https://www.rcsb.org, accessed on 5 May 2020) and UniProt knowledgebase (http://www.uniprot.org/, accessed on 8 May 2020). PBP (PDB code: 6BSQ), SrtA (PDB code: 2KW8), DNA gyrase (PDB code: 4KSG), RNA polymerase subunit alpha (PDB code: 1BDF), ClyM (PDB code: 5DZT), and PgrX (PDB code: 2AW6) were retrieved from Protein Data Bank, while others were retrieved from UniProt. They were MurB (UniProt ID: Q830P3), RNA polymerase subunit beta (UniProt ID: Q82Z41), ribosomal subunit 30S (UniProt ID: Q82ZI6), ribosomal subunit 50S (UniProt ID: Q839E6), FsrB (UniProt ID: G8ADN9), and GBAP (UniProt ID: G8ADP0). Meanwhile, the eight ligands considered were *M. pendans* compounds that were published in previous studies [31,32,33]. They were (**1**) butein or 2′,3,4,4′-Tetrahydroxychalcone, (**2**) biflavonoid, (**3**) 3″-methoxy-epicatechin-3-*O*-epicatechin, (**4**) benzoic acid, (**5**) dibenzo-*p*-dioxin-2,8-dicarboxylic acid, (**6**) pomolic acid, (**7**) betulin, and (**8**) *β*-sitosterol tridecanoil glucopyranose. Moreover, there were fifteen ligands as the positive control, including P or penicillin (CID 2349), Ca or carbapenems (CID 443582), Gly or glycopeptides (CID 56928060), Q or Quercetin (CID 5280343), Ga or gatifloxacin (CID 5379), S or sitafloxacin (CID 461399), M or myxopyronin (CID 136669146), R or Rifamycin (CID 6324616), A or Amoxicillin (CID 33613), Ce or Cefixime (CID 5362065, Cur or Curcumin (CID 969516), AMP (CID 6083), and AA or ambuic acid (CID 11152290) retrieved from PubChem compound database (https://www.ncbi.nlm.nih.gov/pccompound, accessed on 3 May 2020).

### 4.2. Methods

#### 4.2.1. In Silico Characterization of the *M. Pendans* Compounds

The characteristics of compounds **1**–**8** were confirmed using two online software applications. The chemical structures of the four *M. pendans* compounds were converted using a ChemDraw in the CDX file format. The 3D structure of the MOL file for all compounds was retrieved from the PubChem Compound database. Those MOL files were used to convert the chemical structure into a 3D format using OPEN BABEL 2.4.1 program, in PDB file format [41]. The 3D structure model of PBP (PDB code: 6BSQ), SrtA (PDB code: 2KW8), DNA gyrase (PDB code: 4KSG), RNA polymerase, subunit alpha (PDB code: 1BDF), ClyM (PDB code: 5DZT), and PgrX (PDB code: 2AW6) were retrieved from Protein Data Bank. Meanwhile, others were retrieved from UniProt. They are MurB (UniProt ID: Q830P3), RNA polymerase subunit beta (UniProt ID: Q82Z41), ribosomal subunit 30S (UniProt ID: Q82ZI6), ribosomal subunit 50S (UniProt ID: Q839E6), FsrB (UniProt ID: G8ADN9), and GBAP (UniProt ID: G8ADP0), which were built using the SWISS-MODEL server (https://swissmodel.expasy.org/, accessed on 8 May 2020) in PDB file format [42].

#### 4.2.2. PASS Online Analysis of *M. pendans* Compounds

To determine the activity prediction of compounds, *M. pendans* compounds were analyzed with Prediction of Activity Spectra for Substances (PASS) online program, found at http://www.pharmaexpert.ru/passonline/predict.php, accessed on 15 December 2020 [43].

#### 4.2.3. Molecular Docking between Target Protein and *M. Pendans* Compounds

Automated docking studies were performed using AutoDock Vina in the PyRx 0.8 software application [44]. All target proteins (PBP, MurB, SrtA, DNA gyrase, RNA polymerase subunit alpha and RNA polymerase subunit beta, ribosomal subunit 30S, ribosomal subunit 50S, ClyM, FsrB, GBAP, and PgrX) were loaded to become macromolecules. All compounds **1**–**8** from *M. pendans* as ligands were subject to binding to each protein target; the ligands were free for blind docking. The docking process was initiated by selecting the macromolecule and the eight ligands. Position of grid box: PBP (X: 44,6994, Y: 21,1436, Z: 9,0408), MurB (X: −21.554, Y: 32.9988, Z: −4.3529), SrtA (128.612, Y: −2,4687, Z: −24.2904), DNA gyrase (X: 14.4069, Y: 0.7488, Z: 7.6851), RNA polymerase subunit α (X: 47.0271, Y: 15.3509, Z: 32.8666), RNA polymerase-β (X: 177.594, Y: 152.051, Z: 152.6960), FsrB (X: 42.4764, Y: 43.9861, Z: 1.7210), PgrX (X: −39.9609, Y: 36.154, Z: −0.2834), ClyM (X: 36.0740, Y: 29.4745, Y: 85.2444), GBAP (X: 21.3901, Y: 14.4990, Z: 58.0145), ribosomal subunit 30S (X: 328.606, Y: 181.527, Z: 256.9210), ribosomal subunit 50S (X: 290.930, Y: 512.527, Z: 235.9659). Moreover, step-by-step manual instructions were followed until the bonding energy and hydrogen bond of macromolecule-ligand appeared. The selected conformations were conformation with the lowest binding energy, which had a bonding energy score that less than 1.0 Å in positional root-mean-square deviation (RMSD) [45]. This procedure was repeated ten times.

#### 4.2.4. Complex Protein-Ligand Visualization and Analysis

The final step was to analyze docking results using PYMOL, Discovery Studio, and the online program ProteinsPlus [46]. Docking poses and molecular interaction of each protein-ligand complex can be visualized by PYMOL. To show which residues bind to a ligand, the ProteinsPlus program was used to analyze the protein-ligand complex file and then the picture of molecular interactions come out in a 2D structure. For the best visualization, those molecular interactions were illustrated in a 3D molecular picture. The docking poses of each protein–ligand complex were compared to the 3D structure of a protein that bound ligands on the catalytic sites of each protein. It was supposed to evaluate the similarity of the ligation pose of the compound to another compound that bound ligands on that site.

#### 4.2.5. Analysis of Lipinski Prediction

To determine the penetration ability into the cell wall, we conducted the Lipinski rule test through http://www.scfbio-iitd.res.in/software/drugdesign/lipinski.jsp#anchortag, accessed on 1 February 2021 [37].

#### 4.2.6. Analysis of ADMET and Drug-Likeness Prediction 

We predicted ADMET using online software (http://www.swissadme.ch/, accessed on 16 February 2021) by entering a list of SMILES of chemical ligands running the program [47]. Meanwhile, drug-likeness was predicted through OSIRIS Property Explorer (http://www.organic-chemistry.org/prog/peo/, accessed on 16 February 2021) [48].

## 5. Conclusions

According to in silico studies, compounds **1**–**8** were predicted to be able to block proteins that had a role in cell wall biosynthesis more extremely compared to other proteins. However, they inhibited ClyM with the highest average binding affinity. In other words, they had antibacterial activity through quorum sensing disruption. Based on the structure–activity relationship analysis, compound **2** can be a lead compound for the antibacterial agent in each pathway. However, it does not meet Lipinski’s rule and is just a little out of the range, but it is still considered safe for an oral drug. On the other hand, compounds **1**, **4,** and **5** have a binding affinity lower than compound **2**, but they meet the five rules. Therefore, the three compounds can act as candidates for the antibacterial drug. This study acts as a fundamental evaluation for in vitro and clinical studies in future research.

## Figures and Tables

**Figure 1 molecules-26-02465-f001:**
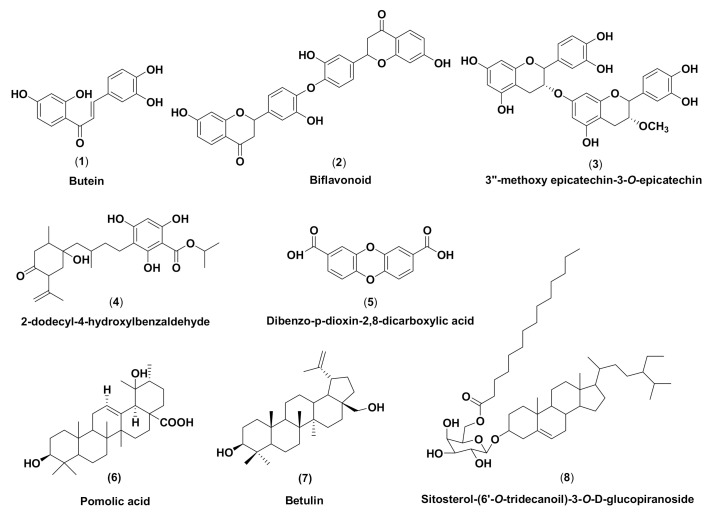
Chemical structures *of M. pendans* compounds: butein (**1**), biflavonoid (**2**), 3″-methoxy epicatechin-3-*O*-epicatechin (**3**), 2-dodecyl-4-hydroxylbenzaldehyde (**4**), 2-dodecyl-4-hydroxylbenzaldehyde (**5**), pomolic acid (**6**), betulin (**7**), and sitosterol-(6′-*O*-tridecanoil)-3-*O*-*β*-D-glucopyranoside (**8**).

**Figure 2 molecules-26-02465-f002:**
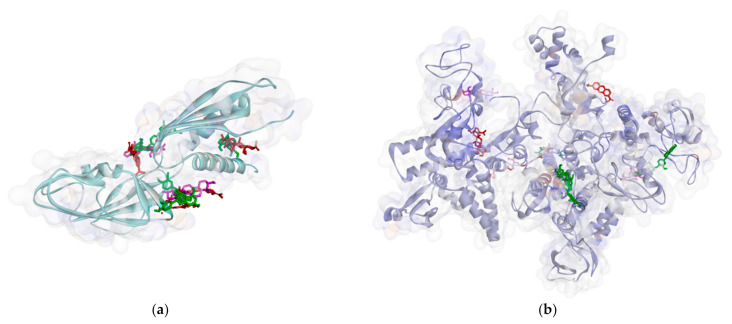
Ligand positions in RNA polymerase alpha (**a**), and RNA polymerase beta (**b**).

**Figure 3 molecules-26-02465-f003:**
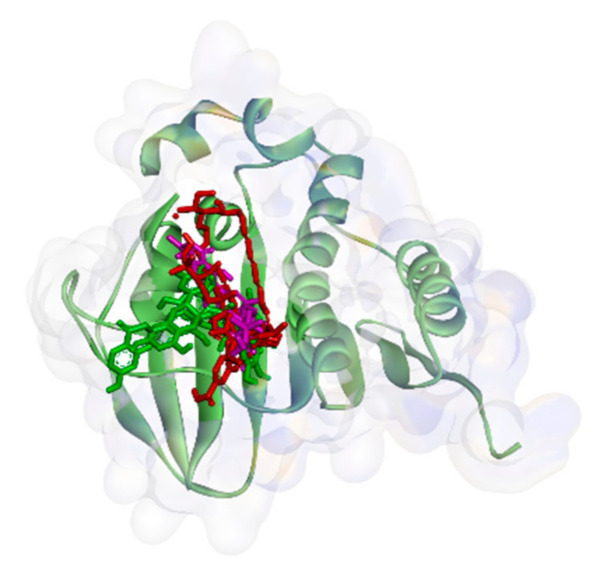
Ligand positions in DNA gyrase.

**Figure 4 molecules-26-02465-f004:**
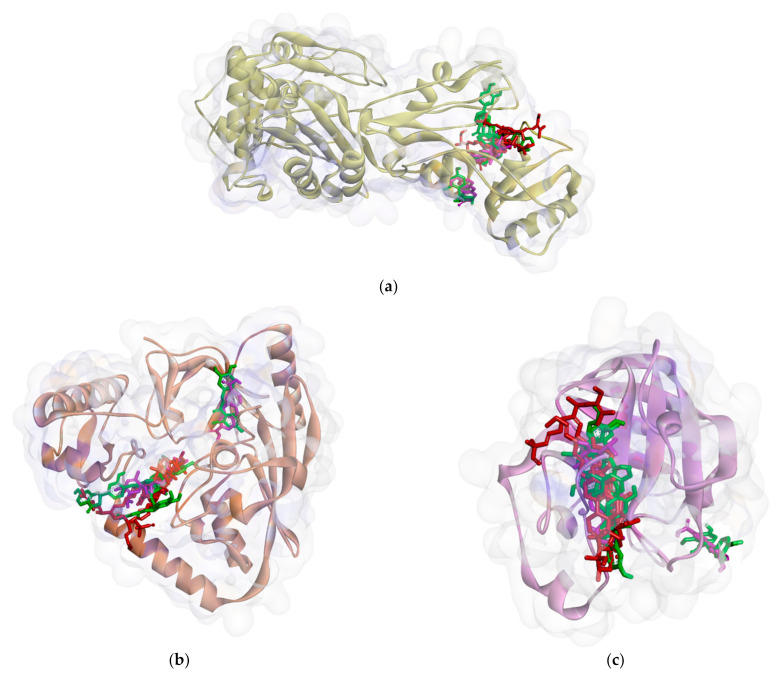
Ligand positions in cell wall protein PBP (**a**), MurB (**b**), and SrtA (**c**).

**Figure 5 molecules-26-02465-f005:**
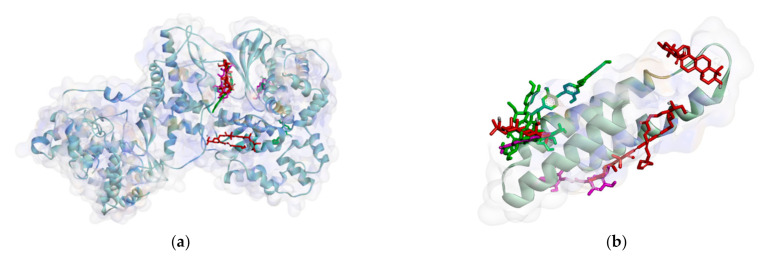
Ligands position in quorum sensing protein: ClyM (**a**), FsrB (**b**), GBAP (**c**), and PgrX (**d**).

**Figure 6 molecules-26-02465-f006:**
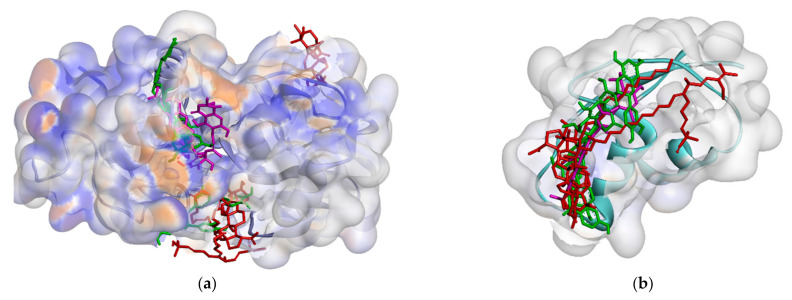
Ligand positions in protein ribosomal subunit 30S (**a**) and ribosomal subunit 50S (**b**).

**Figure 7 molecules-26-02465-f007:**
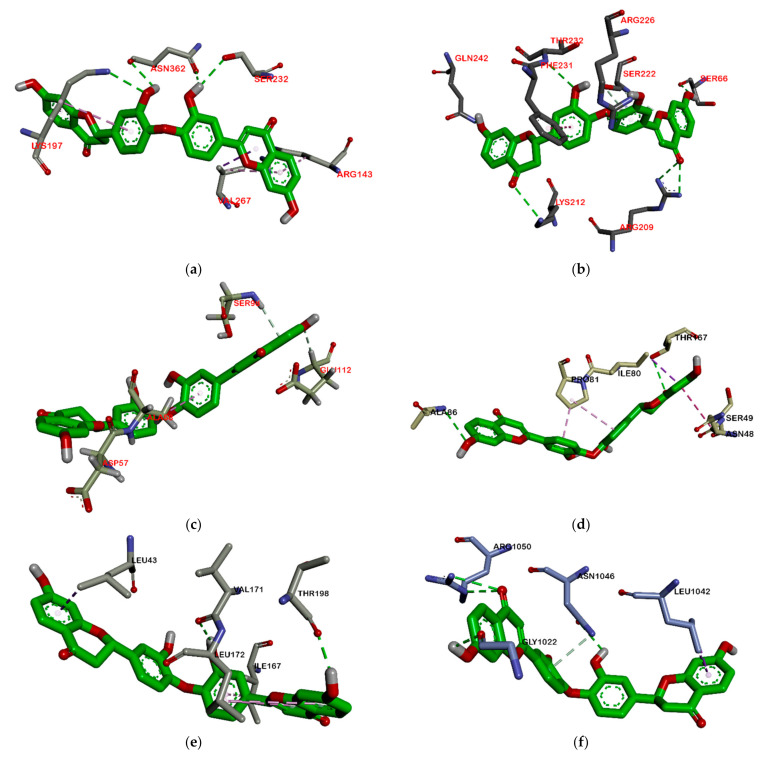
Interaction biflavonoid with PBP (**a**), MurB (**b**), SrtA (**c**), DNA gyrase (**d**), RNA polymerase alpha (**e**), RNA polymerase beta (**f**).

**Figure 8 molecules-26-02465-f008:**
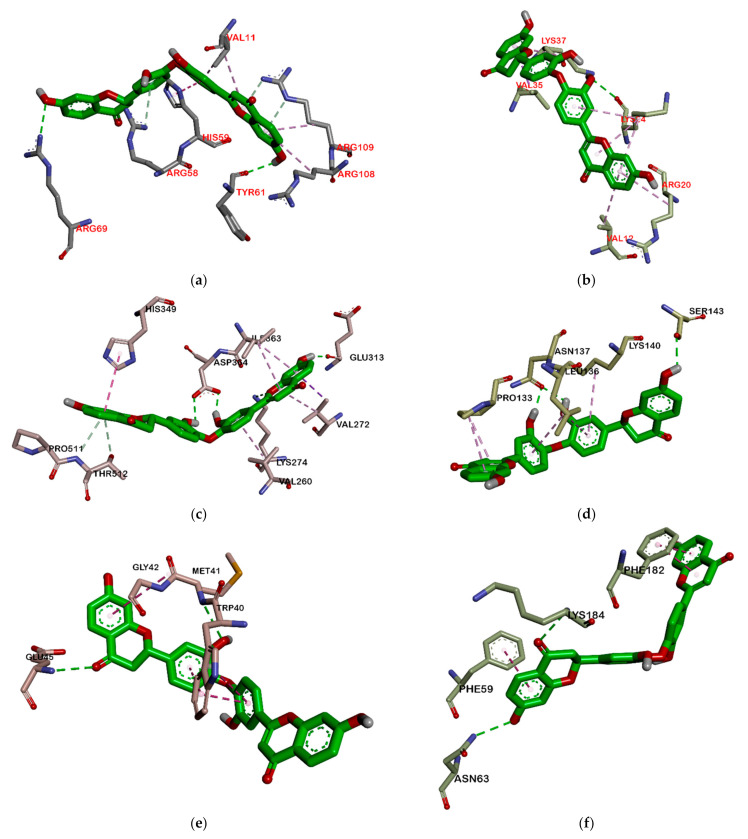
Interaction biflavonoid with protein ribosomal subunit 30S (**a**), ribosomal subunit 50S (**b**), ClyM (**c**), FsrB (**d**), GBAP (**e**), and PgrX (**f**).

**Figure 9 molecules-26-02465-f009:**
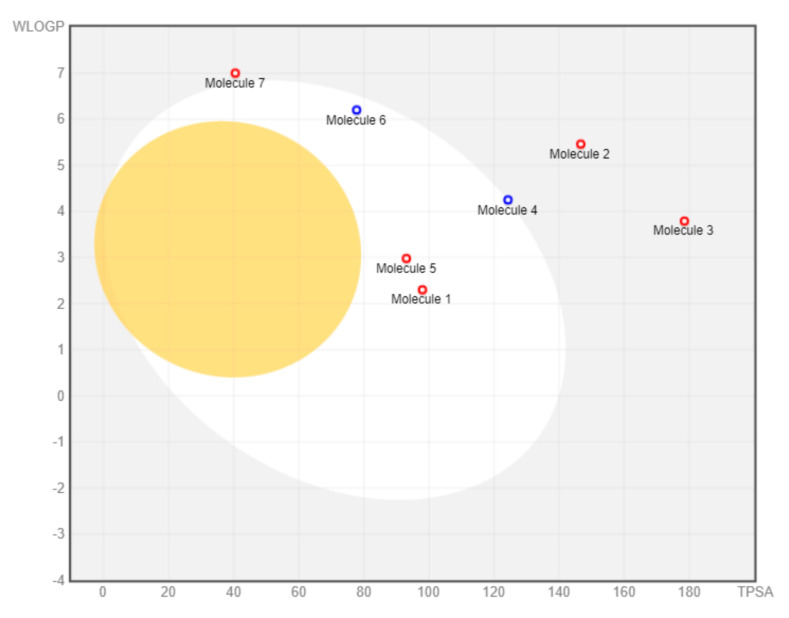
BOILED-Egg visualization.

**Table 1 molecules-26-02465-t001:** Prediction activity of *M. pendans* compounds through PASS analysis.

No	Bioactivity	Prediction to be Active (Pa) of Compound							
		1	2	3	4	5	6	7	8
General									
1	Antibacterial	0.375	0.326	0.344	0.441	-	-	0.242	-
2	Antibiotic	-	0.095	-	0.251	-	-	-	-
3	Antifungal	0.494	0.490	0.498	0.622	-	0.526	0.507	0.327
4	Anti-infective	0.442	0.375	-	-	-	-	-	-
5	Antimycobacterial	0.621	0.347	-	-	-	-	-	-
6	Antiparasitic	0.483	-	-	-	-	-	0.199	-
7	Antiseptic	0.783	0.271	-	0.277	0.751	-	-	-
8	Anti-tuberculosis	0.583	-	-	-	-	-	-	-
DNA Synthesis Pathway									
1	DNA synthesis inhibitor	-	-	-	-	-	-	0.247	0.195
2	DNA-3-methyladenine glycosylase I inhibitor	0.255	-	-	-	0.725	-	-	-
3	DNA-(apurinic or apyrimidinic site) lyase inhibitor	0.255	0.217	-	-	0.829	-	-	-
4	DNA gyrase inhibitor	-	0.033	0.159	-	-	-	-	-
5	DNA ligase (ATP) inhibitor	0.384	0.401	0.386	0.221	-	0.733	0.496	-
6	DNA polymerase I inhibitor	0.260	0.248	0.299	-	-	-	0.360	0.253
7	DNA directed RNA polymerase inhibitor	-	-	0.137	0.164	-	-	0.172	0.338
8	Transcription factor inhibitor	0.511	-	-	-	-	-	-	-
Protein Synthesis Pathway									
1	Protein synthesis inhibitor	-	-	-	0.322	-	-	0.202	0.246
2	Protein 30S ribosomal subunit inhibitor	-	-	0.077	-	-	-	-	-
3	Tpr proteinase (Porphyromonas gingivalis) inhibitor	-	-	-	-	0.862	-	-	-
**No**	**Bioactivity (38)**	**Prediction to be Active (Pa) of Compound**							
		**1**	**2**	**3**	**4**	**5**	**6**	**7**	**8**
Cell Wall Biosynthesis Pathway									
1	Cell wall biosynthesis inhibitor	0.227	-	0.204	-	-	-	-	-
2	Cell adhesion molecule inhibitor	0.327	-	-	-	-	-	-	-
3	Membrane integrity antagonist	0.234	0.516	0.263	-	-	-	0.496	-
4	Membrane permeability inhibitor	0.745	0.788	0.700	0.680	0.820	-	0.411	-
5	Peptidoglycan glycosyltransferase inhibitor	-	0.327	-	-	-	0.460	0.590	-
6	Phospholipid-translocating ATPase inhibitor	-	-	-	-	0.815	-	-	-
7	UDP-glucuronosyltransferase substrate	0.709	0.827	0.771	0.527	-	-	0.554	0.183
8	UDP-N-acetylglucosamine 4-epimerase inhibitor	0.559	-	-	-	0.924	-	-	-
RNA Synthesis Pathway									
1	RNA synthesis inhibitor	0.313	0.326	0.328	0.261	-	-	0.360	0.208
2	RNA directed DNA polymerase inhibitor	0.161	0.172	0.207	0.204	-	-	0.161	-
3	RNA-directed RNA polymerase inhibitor	0.418	-	-	-	-	-	0.276	-
4	tRNA nucleotidyltransferase inhibitor	0.234	-	-	-	0.805	-	-	-
5	tRNA-pseudouridine synthase I inhibitor	0.359	0.217	-	-	0.902	-	-	-
6	Alanine-tRNA ligase inhibitor	0.184	-	-	-	-	-	-	-
7	Aspartate-tRNA ligase inhibitor	0.145	0.196	-	-	-	-	-	-
8	Asparagine-tRNA ligase inhibitor	0.124	0.151	-	-	-	-	-	-
9	Glutamate-tRNA ligase inhibitor	0.248	0.205	-	-	-	-	-	-

**Table 2 molecules-26-02465-t002:** Prediction inactivity of compound Bioactivity from PASS online (Pi).

No	Bioactivity	Prediction to Be Active (Pi)							
		1	2	3	4	5	6	7	8
General									
1	Antibacterial	0.030	0.051	0.045	0.023	-	-	0.087	-
2	Antibiotic	-	0.087	-	0.020	-	-	-	-
3	Antifungal	0.030	0.032	0.031	0.016	-	0.026	0.029	0.071
4	Anti0infective	0.030	0.057	-	-	-	-	-	-
5	Antimycobacterial	0.000	0.056	-	-	-	-	-	-
6	Antiparasitic	0.010	-	-	-	-	-	0.098	-
7	Antiseptic	0.000	0.043	-	0.041	0.005	-	-	-
8	Anti-tuberculosis	0.000	-	-	-	-	-	-	-
DNA Synthesis Pathway									
1	DNA synthesis inhibitor	-	-	-				0.086	0.168
2	DNA-3-methyladenine glycosylase I inhibitor	0.007	-	-	-	0.003	-	-	-
3	DNA-(apurinic or apyrimidinic site) lyase inhibitor	0.010	0.141	-	-	0.004	-	-	-
4	DNA gyrase inhibitor	-	0.015	0.003	-	-	-	-	-
5	DNA ligase (ATP) inhibitor	0.010	0.016	0.019	0.087	-	0.001	0.006	-
6	DNA polymerase I inhibitor	0.120	0.154	0.065	-	-	-	0.025	0.143
7	DNA directed RNA polymerase inhibitor	-	-	0.104	0.065	-	-	0.057	0.013
8	Transcription factor inhibitor	0.010	-	-	-	-	-	-	-
Protein Synthesis Pathway									
1	Protein synthesis inhibitor	-	-	-	0.027	-	-	0.053	0.041
2	Protein 30S ribosomal subunit inhibitor	-	-	0.035	-	-	-	-	-
3	Tpr proteinase (Porphyromonas gingivalis) inhibitor	-	-	-	-	0.002	-	-	-
Cell Wall Biosynthesis Pathway									
1	Cell wall biosynthesis inhibitor	0.100	-	0.148	-	-	-	-	-
2	Cell adhesion molecule inhibitor	0.060	-	-	-	-	-	-	-
3	Membrane integrity antagonist	0.162	0.040	0.139	-	-	-	0.496	-
4	Membrane permeability inhibitor	0.023	0.012	0.039	0.047	0.007	-	0.198	-
5	Peptidoglycan glycosyltransferase inhibitor	-	0.091	-	-	-	0.032	0.009	-
6	Phospholipid-translocating ATPase inhibitor	-	-	-	-	0.004	-	-	-
7	UDP-glucuronosyltransferase substrate	0.014	0.005	0.009	0.025	-	-	0.024	0.110
8	UDP-N-acetylglucosamine 4-epimerase inhibitor	0.041	-	-	-	0.002	-	-	-
RNA Synthesis Pathway									
1	RNA synthesis inhibitor	0.046	0.042	0.039	0.088	-	-	0.028	0.158
2	RNA directed DNA polymerase inhibitor	0.160	0.135	0.082	0.085	-	-	0.159	-
3	RNA-directed RNA polymerase inhibitor	0.041	-	-	-	-	-	0.178	-
4	tRNA nucleotidyltransferase inhibitor	0.040	-	-	-	0.002	-	-	-
5	tRNA-pseudouridine synthase I inhibitor	0.056	0.124	-	-	0.002	-	-	-
6	Alanine-tRNA ligase inhibitor	0.099	-	-	-	-	-	-	-
7	Aspartate-tRNA ligase inhibitor	0.070	0.046	-	-	-	-	-	-
8	Asparagine-tRNA ligase inhibitor	0.064	0.048	-	-	-	-	-	-
9	Glutamate-tRNA ligase inhibitor	0.103	0.160	-	-	-	-	-	-

**Table 3 molecules-26-02465-t003:** Binding affinity of the compound against proteins (kcal·mol^−1^).

No.	Compound	PBP	MurB	SrtA	DNA Gyrase	RNA Polymerase		Ribosomal		ClyM	FsrB	GBAP	PgrX
						Alpha	Beta	30S	50S				
1	Compound **1**	−6.9	−7.2	−5.6	−6.6	−6.6	−6.7	−7.5	−5.6	−8.3	−5.8	−5.5	−6.6
2	Compound **2**	−11.2	−11.5	−7.6	−8.6	−8.6	−9.0	−9.4	−6.8	−10.4	−7.7	−6.9	−8.5
3	Compound **3**	−10.5	−9.5	−7.0	−8.3	−8.4	−9.0	−7.4	−5.8	−9.4	−7.1	−6.3	−9.0
4	Compound **4**	−8.0	−7.9	−6.5	−7.1	−5.6	−6.7	−6.1	−4.7	−7.8	−6.0	−5.3	−6.3
5	Compound **5**	−7.5	−8.4	−6.4	−7.2	−6.3	−7.3	−7.5	−5.4	−8.5	−5.8	−5.2	−6.7
6	Compound **6**	−10.1	−8.8	−7.1	−7.5	−7.9	−8.2	−7.6	−6.2	−8.9	−6.6	−6.0	−7.4
7	Compound **7**	−7.5	−6.4	−6.7	−8.1	−7.0	−7.5	−7.0	−6.1	−7.9	−6.1	−5.3	−6.4
8	Compound **8**	−8.1	−6.8	−5.4	−6.0	−5.7	−6.6	−5.9	−5.3	−8.3	−6.7	−4.6	−6.4
Average binding affinity		−8.7	−8.3	−6.5	−7.3	−7.0	−7.6	−7.3	−5.7	−8.7	−6.5	−5.6	−7.2
9	Penicillin	−7.5	NT	NT	NT	NT	NT	NT	NT	NT	NT	NT	NT
10	Carbapenems	−8.5	NT	NT	NT	NT	NT	NT	NT	NT	NT	NT	NT
11	Glycopeptides	NT	−7.4	NT	NT	NT	NT	NT	NT	NT	NT	NT	NT
12	Quercetin	NT	−8.1	NT	NT	NT	NT	NT	NT	NT	NT	NT	NT
13	Amoxicillin	NT	NT	−5.8	NT	NT	NT	NT	NT	NT	NT	NT	−6.3
14	Cefixime	NT	NT	−5.2	NT	NT	NT	NT	NT	NT	NT	NT	−6.9
15	Curcumin	NT	NT	−5.7	NT	NT	NT	NT	NT	NT	NT	NT	NT
16	Sitafloxacin	NT	NT	NT	−6.0	NT	NT	NT	NT	NT	NT	NT	NT
17	Rifamycin	NT	NT	NT	NT	−6.7	−7.7	NT	NT	NT	NT	NT	NT
18	Tetracycline	NT	NT	NT	NT	NT	NT	−7.4	NT	NT	NT	NT	NT
19	Chloramphenicol	NT	NT	NT	NT	NT	NT	NT	−4.9	NT	NT	NT	NT
20	(+)-AMP	NT	NT	NT	NT	NT	NT	NT	NT	−8	NT	NT	NT
21	Ambuic acid	NT	NT	NT	NT	NT	NT	NT	NT	NT	−5.3	−5.3	NT

Note: penicillin-binding protein (PBP), Sortase A (SrtA), Cytolysin M (ClyM), gelatinase-binding activating pheromone (GBAP). NT: not tested.

**Table 4 molecules-26-02465-t004:** Lipinski’s rule result.

Ligand	Lipinski’s Rule of Five					Drug-Likeness
	Molecular Mass (Dalton)	Hydrogen Bond Donor	Hydrogen Bond Acceptors	Log P	Molar Refractivity	
	Less Than 500 Dalton	Less Than 5	Less Than 10	Less Than 5	40–130	Lipinski’s Rule Follows
M1	272	4	5	2.405	72.908	Yes
M2	524	4	9	5.714	137.134	No
M3	576	-	11	4.441	146.781	No
M4	448	4	7	4.249	120.748	Yes
M5	272	2	6	3.138	68.719	Yes
M6	472	3	4	6.204	134.071	No
M7	442	2	2	6.997	132.062	No
M8	772	3	7	10.711	221.185	No

**Table 5 molecules-26-02465-t005:** Drug likeness prediction using OSIRIS Property Explorer.

Ligand	clogP	Solubility	TPSA	Mutagenic	Tumorigenic	Irritant	Reproductive Effective
M1	1.92	−2.66	97.99	High risk	low risk	Medium risk	low risk
M2	4.92	−6.77	142.7	low risk	low risk	low risk	low risk
M3	4.6	−3.41	178.5	low risk	low risk	low risk	low risk
M4	4.87	−4.2	124.2	low risk	low risk	low risk	low risk
M5	2.67	−5.15	93.06	Medium risk	low risk	low risk	low risk
M6	5.18	−5.66	77.76	low risk	low risk	low risk	low risk
M7	6.72	−6.3	40.46	low risk	low risk	low risk	low risk
M8	11.5	−9.75	105.4	low risk	low risk	low risk	low risk

**Table 6 molecules-26-02465-t006:** Pharmacokinetic values of *M. pendans* compounds using SwissADME.

Ligand	GI Absorption	BBB Permeant	Pgp Substrate	Inhibitor of					Bioavailability Score
				CYP1A2	CYP2C19	CYP2C9	CYP2D6	CYP3A4	
M1	High	No	No	Yes	No	Yes	No	Yes	0.55
M2	Low	No	No	No	No	Yes	No	No	0.55
M3	Low	No	No	No	No	No	No	Yes	0.17
M4	Low	No	Yes	No	No	No	No	Yes	0.55
M5	High	No	No	Yes	No	No	No	No	0.56
M6	High	No	Yes	No	No	No	No	No	0.56
M7	Low	No	No	No	No	No	No	No	0.55
M8	Low	No	Yes	No	No	No	No	Yes	0.17

## Data Availability

Not applicable, the study did not report any data.

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
