# Peer review of "Effectiveness of Bioactive Compound as Antibacterial and Anti-Quorum Sensing Agent from *Myrmecodia pendans*: An In Silico Study"

_molecules, 2021, doi:10.3390/molecules26092465_

Round 1

Reviewer 1 Report

The manuscript has been revised according to the remarks and in my opinion, is the paper is ready to be published.

Reviewer 2 Report

  • In my opinion, the paper requires extensive English correction. There are so many errors that it's hard to mention them all in the review. The paper is also a bit chaotic. The subject matter itself is interesting, but the way of presentation makes it unattractive for the reader.
  • The section “Molecular docking between target protein with M. pendans compounds requires more details. How many times docking has been repeated? Have you tested docking with a different initial configuration? The authors wrote “Moreover, step by step was followed as manual instruction until calculating the bonding energy and hydrogen bond of macromolecule-ligand appeared.” What instruction did the authors follow?
  • The authors described the hydrogen bonds and hydrophobic contact between the studied compounds and the selected proteins. Have the authors investigated the conformational changes of the proteins after binding to the ligands?
  • Is the binding affinity shown as the free binding energy? Please verify.

Minor:

In the unit kcal.mol-1 please use „·” instead of „.”,

The references in the text should be given in the following manner: [9,10] or [21-24] instead of [9], [10] and [21]–[24],

Figure 1 with the structures of the compounds should be moved before all the tables.

Reviewer 3 Report

Although a good research idea, the paper presented is not fit for publication in its present form.

  1. Consider re-formulating the abstract. It is a bit to long and contains some specific data (e.g. values for binding affinity) that are hard to digest in this step.
  2. Throughout the manuscript a lot of paragraph would be better presented just as tables (e.g. line 106-113)
  3. The Discussions section does not link with similar research. The results are not compared with similar research or with in vivo results obtained by other research groups that tested the target compounds.
  4. However, the biggest setback for this manuscript has to be the use of the English language. Although sometimes the text is not necessarily flawed from the point of view of grammar, it is not the correct way to present scientific results. Most of times the text is hard to read. The word and phrasing chosen are often not the appropriate one. The text must be completely re-written by a native speaker with some background in scientific writing!

Round 2

Reviewer 2 Report

The manuscript has been revised according to the remarks and  is now ready for publication.

Author Response

Thank you for your valuable appreciation for revised manuscript

Reviewer 3 Report

Although the authors have provided some improvements, the English style is far from that of a scientific article. As such, I still feel this paper needs to be revised by a native speaker, with a scientific background. 

Author Response

Thank you for your suggestion

I try to recheck and rerevise the manuscript carefully

This manuscript is a resubmission of an earlier submission. The following is a list of the peer review reports and author responses from that submission.

Round 1

Reviewer 1 Report

  • The English language and spelling require major revision.
  • Reference or web address for PASS program is missing.
  • meaning of abbreviations of protein should be include at the first mention including as a foosnote under the Table 2, 3,4, 7, 8
  • Binding affinity should has a unit
  • in silico is write italic
  • line 81: The sentence: „The in silico method is a current tool that can show the prediction of the activity of a compound“ is not correct. In silico is not method, it is group of computational methods such as QSAR, molecular docking, molecular simulation etc. Methods do not show the prediction. The activities of compounds could be predicted by computational methods.
  • line 101: Biflavonoid should be biflavonoid

Reviewer 2 Report

The manuscript entitled "Potency of Bioactive Compound as Antibacterial and Anti-Quorum Sensing Agent from Myrmecodia pendans: In silico study" presented for review concerns the evaluation of the activity of naturally-occurring compounds from Myrmecodia pendans, a herb Indonesian plant. The search for new antimicrobial agents is fully justified, but their activity must be supported by appropriate results. Unfortunately, I get the impression that the research in the paper has not been carefully thought out. The results are not adequately supported by the research. Moreover, the research methodology raises my doubts, perhaps because it is poorly described, as are the results. The paper is difficult to understand. I am not a native speaker, but in my opinion an extensive English correction is required. In summary, the manuscript is not suitable for publication in Molecules.

Reviewer 3 Report

The authors' manuscript deals with the in silico analysis of components isolated from Myrmecodia pendans, a medicinal plant to which various effects have been attributed. In particular, for this work, previous studies have demonstrated antibacterial activity of extracts of this plant. The in silico analysis consisted of using the PASS online chemoinformatics tool that predicts the compounds' potential bioactivities and molecular docking on some biological targets that emerged from the chemoinformatic analysis.

Among the strengths that can be highlighted in the manuscript is the combination of chemoinformatic analysis + molecular docking, which is one of the strategies commonly used in computational studies. However, the manuscript's writing is confusing, which makes it difficult to appreciate the work correctly. For this reason, I think that the manuscript is not ready to be published in its present form. Besides, the reviewer has some additional concerns that prevent for now suggesting acceptance of the article:

1) The PASS online tool offers not only the Pa value (probability to be active) but also de Pi value (probability "to be inactive"), which could also be interesting for analysis since most of the Pa values reported looks relatively low. The use of a control substance with known activity (as the authors did in the molecular docking studies) might help understand if the Pa values are high or low. Did additional potential bioactivities were found during the PASS analysis?

2) Tables 3 to 8 could be part of Supplementary information. Instead, I recommend that the authors take advantage of Figures 2 to 6 to illustrate the compounds' interactions with higher theoretical affinity with the analyzed biotargets.

3) The description of the methodology seems incomplete. The authors state that they downloaded the biotargets 3D structure from SWISS-MODEL. Could the authors be more specific on which structures they selected? For example, which accession codes in SWISS-MODEL or PDB codes are associated with each structure? 

4) Could the authors mention which searching sites they selected for the docking study? Was it a local docking study on the active site or a global docking study?

5)The authors have recently reported the in silico and experimental in vitro antibacterial activity of biflavonoid 2 (DOI10.2174/1386207323666200628111348). How do the results of the present study complement the experimental data previously obtained?